# Cost-Performance Evaluation of a Recognition Service of Livestock Activity Using Aerial Images

**Darío G. Lema** [1,*][iD], **Oscar D. Pedrayes** [1][iD], **Rubén Usamentiaga** [1][iD], **Daniel F. García** [1][iD] and **Ángela Alonso** [2][iD]

[1] Department of Computer Science and Engineering, Campus de Viesques, University of Oviedo, 33204 Gijón, Asturias, Spain; UO251056@uniovi.es (O.D.P.); rusamentiaga@uniovi.es (R.U.); dfgarcia@uniovi.es (D.F.G.)

[2] Department of Spatial Data, Seresco S.A., Matemático Pedrayes 23, 33005 Oviedo, Asturias, Spain; angela.alonso@seresco.es

\* Correspondence: UO243567@uniovi.es

**Abstract:** The recognition of livestock activity is essential to be eligible for subsides, to automatically supervise critical activities and to locate stray animals. In recent decades, research has been carried out into animal detection, but this paper also analyzes the detection of other key elements that can be used to verify the presence of livestock activity in a given terrain: manure piles, feeders, silage balls, silage storage areas, and slurry pits. In recent years, the trend is to apply Convolutional Neuronal Networks (CNN) as they offer significantly better results than those obtained by traditional techniques. To implement a livestock activity detection service, the following object detection algorithms have been evaluated: YOLOv2, YOLOv4, YOLOv5, SSD, and Azure Custom Vision. Since YOLOv5 offers the best results, producing a mean average precision (mAP) of 0.94, this detector is selected for the creation of a livestock activity recognition service. In order to deploy the service in the best infrastructure, the performance/cost ratio of various Azure cloud infrastructures are analyzed and compared with a local solution. The result is an efficient and accurate service that can help to identify the presence of livestock activity in a specified terrain.

**Keywords:** livestock activity recognition; Azure; cloud service deployment and cost-performance evaluation; aerial images; CNNs; YOLOv2; YOLOv4; YOLOv5; SSD; Azure Custom Vision





## 1. Introduction

Confirming the presence of livestock activity is necessary to be eligible for a series of subsidies. In the European Union, the most important subsidy is the CAP (Common Agricultural Policy), which in 2019 totaled € 57,980,000,000 [1]. In order to qualify for the subsidy, it is necessary to declare which land the agricultural or livestock activity is being carried out on. It is crucial to create a service to automate this task since, currently, it is very common for an operator to have to go on site to check whether a piece of land complies with the declaration made. Sometimes, drone images are used, which are manually reviewed by an operator. This manual process is very costly due the large surface area to be checked: the European Union has 4 million km$^2$. This work will help to identify the presence of livestock activity, reducing the cost and increasing the speed with which the declaration is confirmed.

The use of computer vision is widespread in all kinds of fields, such as steel monitoring [2], agriculture [3,4] as well as for animal recognition tasks. In order to detect this activity, several studies using Support Vector Machines (SVM) have been carried out [5,6]. SVM is a widely used technique, but, in recent years, it has been displaced by the use of Convolution Neuronal Networks (CNN), due to the good results offered. The advantage of using CNNs instead of SVM is the possibility of using GPUs instead of CPUs for training these object detection algorithms. The popularization of GPUs has allowed the development of several object detectors based on CNNs such as R-CNN [7], SSD [8], or YOLO [9].

To compare these object detection algorithms, it is necessary to establish a metric. The most widespread is average precision, AP [10].

The vast majority of CNNs are evaluated on generalist datasets, usually Pascal VOC [10] and COCO [11]. The use of these datasets to compare object detection algorithms is correct, due to the large number of images and annotations they contain, but the images used are not based on aerial imagery. In order to carry out livestock activity detection in a practical way, it is necessary to use aerial images, since it is necessary to cover a lot of ground. Thus, results obtained by means of non-aerial images are not conclusive for this work. In [12–14], different CNNs have been tested using aerial imagery datasets. The main problem with these works is that their aim is usually to detect objects of large dimensions, such as airplanes and ships. These objects, despite being smaller (in terms of number of pixels) than those used in non-aerial imagery datasets, are still much longer than the objects of interest for the recognition of livestock activity, in which the objects may have dimensions of a few pixels.

Two types of datasets are used in works related to animal detection: high resolution [5,15,16], where the Ground Sampling Distance (GSD) is low, and low resolution [6,17–19], where the GSD is large. In both cases, drone (Unmanned Aerial Vehicle or UAV) images are usually chosen over satellite images, since the objects of interest often have barely one pixel. It seems obvious that the lower the resolution, the more difficult it is to detect objects.

SVM is used in [5] to detect and count cows; therefore, a suitable aerial dataset is created to perform this task. For cow detection, SVM is used, with an AP of 0.66. Other works, such as [6], use SVM, to detect large mammals in the African savannah. In this case, although a high recall rate is obtained, it is necessary for an operator to review the detections. SVM was widely used in the past but has now been displaced by CNNs due to the increased accuracy offered. A method based on Faster R-CNN [20] is used in [21] to detect dairy goats. This new method offers better results than the classic Faster R-CNN and deals with one of its main problems: time-consuming procedure. To detect dairy goats in a surveillance video, accuracy is necessary but also low inference time. The proposed method not only achieves good results (AP = 0.92), but is also twice as fast as the classic Faster R-CNN. Faster R-CNN is also used in [22] with good results (AP = 0.92) in detecting kiangs (a wild member of the horse family native to Tibet). The conclusion reached in this work is that feature stride shortening and anchor box size optimization improve the detection of small animals. YOLOv2 [23] is another state-of-the-art object detector algorithm widely used for animal detection. In [19], its final goal is to identify individual Holstein Friesian cows by their coat pattern. For this purpose, three CNNs are running on board an aircraft. This system is composed of YOLOv2 as the object detector, an exploratory agency CNN to calculate the best route to follow, and an InceptionV3-based biometric network for individual cattle identification. The system reaches an accuracy of 91.9%. In addition to object detectors, instance segmentation is also used for animal detection. In [16], Mask-RCNN [24] is used, reaching an accuracy of 96% in livestock classification and 91.2% in cattle and sheep counting.

The main objective of this work is to contribute to the recognition of livestock activity. Consequently, the following contributions have also been made:

- Recognition of livestock activity. The vast majority of previous works do not address livestock activity recognition, but focus only on the detection of animals, whether wild or farm animals. This article not only focuses on the presence of animals, but also evaluates other elements present on any farm: manure piles, silage balls, feeders, silage storage areas, and slurry pits. By recognizing these objects, evidence of the presence of livestock activity is obtained; therefore, it is not necessary to analyze the possible movement patterns of these objects.
- Evaluation of the most recent object detection algorithms that have not been used in the agricultural field. To detect the key objects as accurately as possible, it is necessary to evaluate several object detectors. Previous works have used the object

detectors considered as state of the art, but other more innovative detectors such as Azure Custom Vision [25], YOLOv4 [26] or YOLOv5 [27] have not yet been evaluated. In this work, these new object detectors are analyzed and compared with more traditional detectors such as YOLOv2 [23] and SSD [8], considered state of the art just until recently.

- Use of geo-referenced images. The normal pipeline is to use one image as input, do inference, apply Non-Maximal Suppresion (NMS) and finally get the bounding boxes in pixels. This is not enough to prove the presence of livestock activity. By using geo-referenced images after inference and NMS, it is possible to translate the pixels of the bounding boxes into georeferences. This way, it is possible to geo-reference objects considered proof of in livestock activity recognition. It is important to measure the time needed to translate pixels from bounding boxes to georeferences, as this task affects the performance of the service.

- Analysis of the performance/cost ratio for each of the infrastructures evaluated. In order to deploy the service, it is necessary to evaluate different infrastructures, in order to select the best. This work evaluates both a local infrastructure and the following cloud infrastructures offered by Azure: Infrastructure as a Service (IaaS), Container as a Service (CaaS), and Function as a Service (FaaS). In order to select the best service, both time and cost are important. This paper studies the performance/cost ratio of the evaluated infrastructures. Normally, this issue is ignored as only the performance metrics of the model are evaluated, without taking into account the deployment, which is of vital importance for a service in production.

This paper details the process carried out to develop the livestock activity recognition system. This process is shown in Figure 1. Firstly, with the help of a drone, a large number of images are obtained. These images are duly annotated so that they can be used with different object detectors. Once the different object detectors have been analyzed, it is concluded that YOLOv5 offers the best results. Finally, the system in charge of recognizing livestock activity in a local and cloud environment is created and tested.

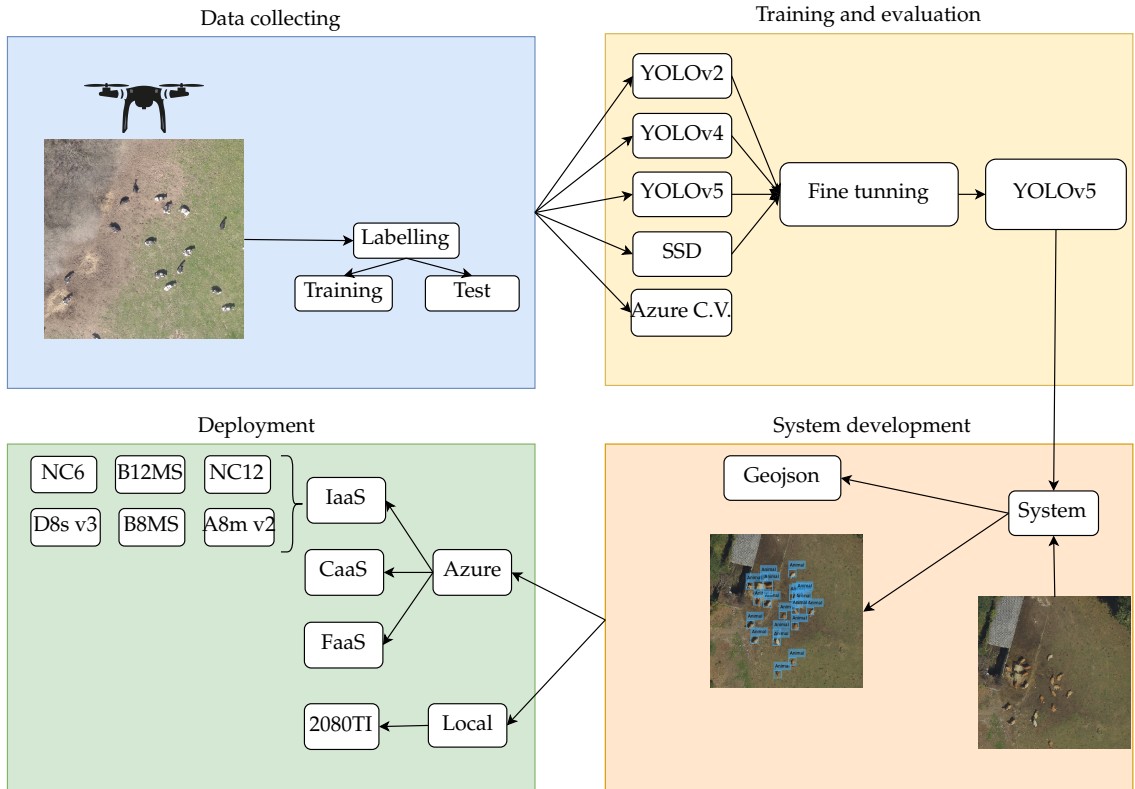

**Figure 1.** Workflow for the development of the livestock activity recognition system.

## 2. Materials and Methods

Different object detection algorithms are examined in this section, together with the dataset used.

### *2.1. Analysis of Object Detection Algorithms*

2.1.1. YOLOv2

YOLOv1 [9] is the first object detection algorithm in the You Only Look Once family. YOLO works as follows: the image is divided into an $S \times S$ grid, with each cell responsible for predicting the objects whose center falls in that cell. Each of these cells predicts $B$ bounding boxes, along with their confidence scores. At the same time, $C$ conditional class probabilities are predicted in each cell. YOLOv1's architecture is inspired by GoogLeNet's [28], using 24 convolutional layers and two fully connected layers. Its main advantage over other object detection algorithms is its inference speed. YOLOv1 runs at 45 FPS, much faster than other object detection algorithms, such as Faster RCNN [20], but its main drawback is that it is not as accurate. For this reason, YOLOv2 [23] was created. YOLOv2 introduces the Darknet-19 backbone to calculate feature maps and removes the fully connected layers. YOLOv2 introduces the concept of anchor boxes in the YOLO family unlike YOLOv1. In YOLOv2, it is not necessary to generate bounding boxes from scratch. Most of the objects in a dataset have similar shapes and aspect ratios, which can be taken advantage of. After some initial guesses, it is only necessary to calculate offsets from these anchor boxes. In other detectors, the anchor boxes are picked by hand, but, in YOLOv2, they are calculated using k-means [23]. Not selecting anchors by hand results in better initial guesses and thus in better outcomes. All these changes produce more accurate detections without affecting the speed of YOLOv2, as it runs at 67 FPS with $416 \times 416$ input size.

### 2.1.2. YOLOv4 and YOLOv5

YOLOv2 is an improvement on YOLOv1, but it still struggles with small objects. This is important because, in livestock recognition, most of the objects to be detected are small, making YOLOv3 [29] a significant development. The main contribution of YOLOv3 to this work is that it is able to make predictions at three different scales, improving predictions for small objects. In addition to three-scale prediction, YOLOv3 uses Darknet-53 [29], a variant of Darknet-19, composed of 53 convolutional layers. This makes YOLOv3 slower than YOLOv2, running at 34.5 FPS. YOLOv3 also uses anchor boxes, three for each scale, meaning that, for the same image, it produces more bounding boxes than YOLOv2.

After YOLOv3, YOLOv4 [26] and YOLOv5 [27] were released. Like current state-of-the-art detectors, these two new versions of YOLO are composed of three parts:

- Backbone: CSPDarknet-53.
- Neck: PA-NET.
- Head: YOLOv3.

The main contribution of YOLOv5 is to integrate new ideas from other works, achieving high accuracy and maintaining real-time performance. Some of these new ideas are related to data augmentation. In YOLOv5, the technique known as mosaic augmentation is used, whereby four images are combined into one. To do this, four slices of four different images are generated and merged into a new image. In this way, classes that would otherwise never appear in the same image can be combined. Another idea introduced in YOLOv5 is to use auto anchor boxes. The anchor boxes are still obtained from the k-means clustering algorithm, but now no configuration is required, since, in the event that the specified anchor boxes are not the correct ones, new anchor boxes appropriate to the dataset in question are automatically calculated.

### 2.1.3. SSD

SSD or Single Shot Multibox Detector [8] arose due to the need for a detector that improves the results of YOLOv1 and is still fast, as Faster R-CNN runs at 7 FPS. Object

detectors can be divided into two categories: one-state or two-state. Two-state detectors first propose a number of regions of interest in each of the images, and then classify each of these regions. Two-state detectors produce good results, but, due to the large number of regions to process, they are very time-consuming. R-CNN and its variants are examples of two-state detectors. SSD and YOLO are one-state detectors, performing detections by regression on the proposed bounding boxes.

The network is trained with a weighted sum of localization and confidence loss, using hard negative mining and data augmentation. The SSD version used has an input size of $300 \times 300$. Therefore, either all the images must be resized from $416 \times 416$ to $300 \times 300$, or new $300 \times 300$ crops must be made (see Section 2.2). After evaluating both options, it was concluded that the second option is the most valid because resizing images would reduce resolution making comparison with other object detection algorithms impossible. For this reason, new $300 \times 300$ crops were made for SSD.

### 2.1.4. Azure Custom Vision

Azure Custom Vision [25] is a service provided by Microsoft that can be applied to classification and object detection tasks. In this work, Azure Custom Vision for object detection is evaluated.

Azure Custom Vision operation is very simple: the first step is to upload the dataset with images and annotations in the correct format. After the dataset is uploaded, it is necessary to train the model. In Azure Custom Vision, no hyperparameter needs to be tuned, as the best configuration is automatically and transparently set. Finally, after training the model, predictions can be made.

Azure Custom Vision has the advantage of generating models very quickly, as the training takes only a few minutes. Its main drawback is that it is not free, unlike other object detection algorithms. Table 1 shows the three costs of Azure Custom Vision: storage, training, and predictions. As it is only necessary to upload training images, the process followed to calculate the offered metrics is not known. To overcome this issue, the tool described in [30] is used to calculate compatible performance metrics.

**Table 1.** Azure Custom Vision storage, training and prediction costs.

| Concept | Cost |
|---|---|
| Storage (cost per month) | € 0.59 /1000 images |
| Training | € 16.86 /h |
| Predictions | € 1.69 /1000 images |

### 2.2. Dataset

### 2.2.1. Data Collection

The dataset used is composed of 50 large images (about $10,000 \times 10,000$ pixels). All these images contain relevant objects of livestock activities developed in the northwest of Spain: animals, feeders, manures piles, silage balls, silage storage areas, and slurry pits. Figure 2 shows two samples of each class. A drone equipped with Micasense ALTUM and Alpha 7 cameras was used to collect these images. After all the images were collected, they were annotated. As for semantic segmentation, it is necessary to classify every single pixel of the image, which is a very time-consuming task, and it has been decided to use object detection, where only the bounding boxes are needed. The annotation format used is the following: class_id x_center y_center width height, normalizing all coordinates between 0 and 1. After this process, $416 \times 416$ crops were created, obtaining the number of images and objects shown in Table 2.

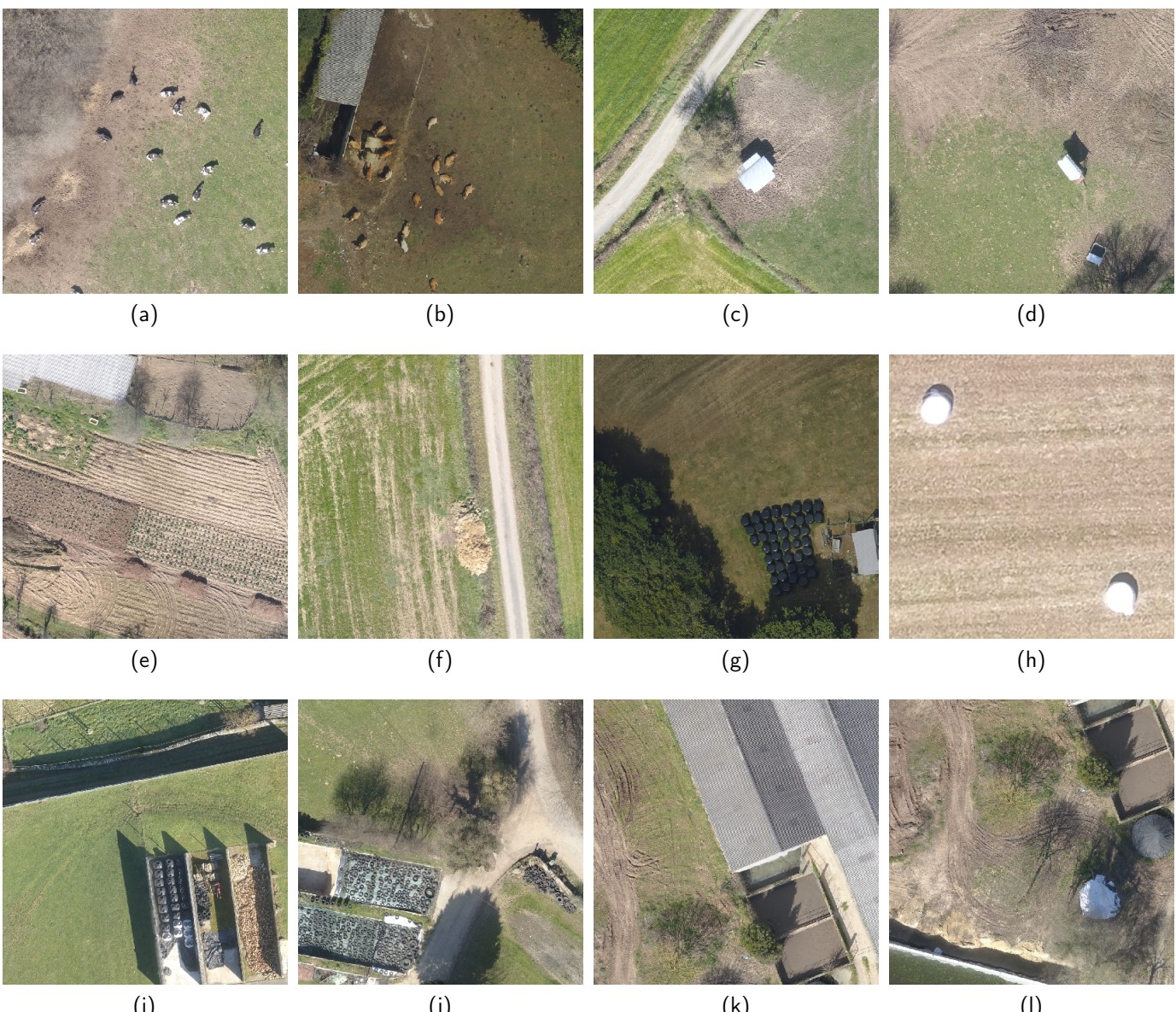

**Figure 2.** Example images of all classes taken from the proposed dataset. (**a**,**b**) animals; (**c**,**d**) feeders; (**e**,**f**) manure piles; (**g**,**h**) silage balls; (**i**,**j**) silage storage areas; (**k**,**l**) slurry pits.

**Table 2.** Original number of images and instances per class after cropping original images to $416 \times 416$.

| Class | Number of Images | Number of Objects |
|---|---|---|
| Animal | 278 | 768 |
| Feeder | 73 | 88 |
| Manure pile | 23 | 69 |
| Silage ball | 135 | 192 |
| Silo S. A. | 9 | 10 |
| Slurry pit | 25 | 26 |

2.2.2. Analysis of the Dataset

The number of images and objects is too low to create a robust model. For this reason, the following augmentation technique was applied to get more data. Firstly, the dataset was divided into eight subsets with the same number of images (four for training and four for testing). This division makes possible to carry out cross testing. Secondly, for the

training subsets, a new image was generated from the objects already annotated. To do this, the new image was centered on one of these objects, and a 416 × 416 crop was generated. In the new crop, there is always an object in the center of the image, with additional objects around it. If an object is cut off, it is taken into account if at 50% or more of the object is present in the new image. After applying this technique, the number of images and objects increases considerably, as shown in Table 3. Despite the augmentation of the data, the dataset is not balanced since, as shown in Table 4, the number of silage storage areas and slurry pits in the test set is very low (the augmentation technique is only applied on the training set).

According to Figure 3a,b, there is little intra-class size variability, with the exception of classes silage storage area and slurry pit. The animal class is the largest, with 7792 instances (see Table 3), with its median for width and height being 20 and 20 pixels, respectively, which can be translated into a bounding box of 1.60 × 1.60 m. Silage storage area is the least represented class with, 31 objects. Its median for width and height is 176 and 146 pixels, respectively. Low intra-class variability facilitates the detection of objects, as high intra-class variability would require detectors to be flexible enough to handle objects of different sizes. Furthermore, there are no significant differences between the width and height of the objects in a single class.

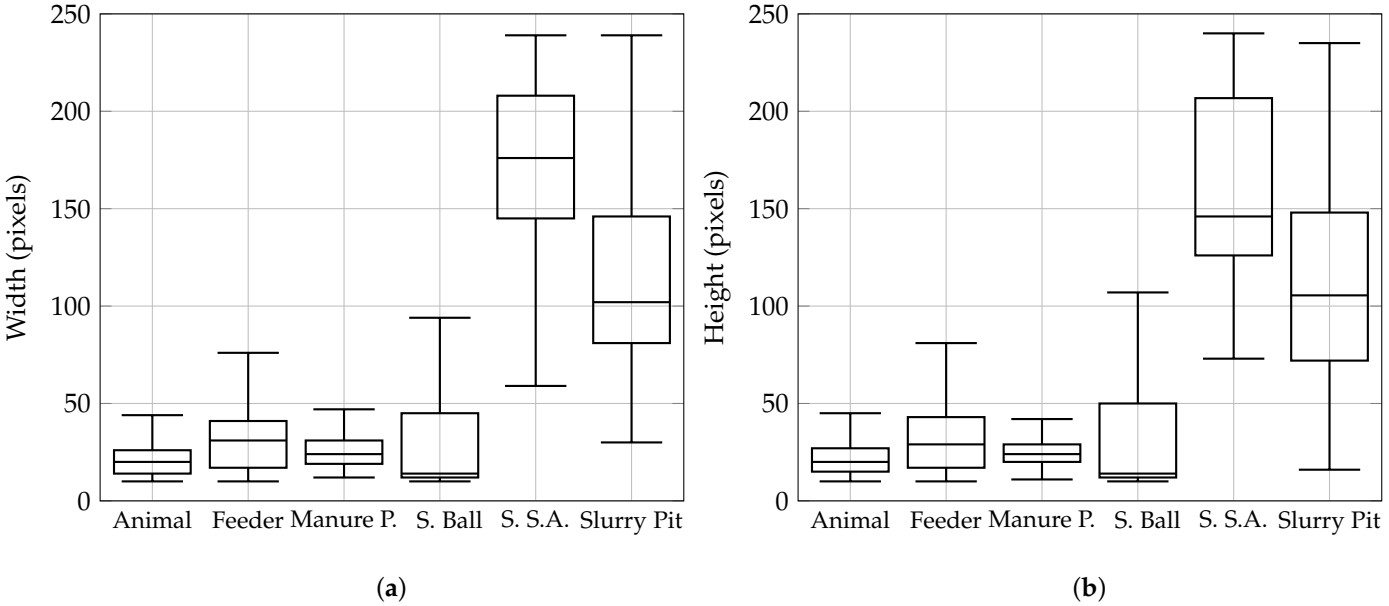

**(a)**                                                                                               **(b)**

**Figure 3.** Object distribution per class of the augmented dataset. (**a**) analysis of the width of each class; (**b**) analysis of the height of each class.

**Table 3.** Total number of images and instances used per class after applying an augmentation technique.

| Class | Number of Images | Number of Objects |
|---|---|---|
| Animal | 1139 | 7792 |
| Feeder | 203 | 264 |
| Manure pile | 99 | 817 |
| Silage ball | 394 | 1065 |
| Silage S.A. | 25 | 31 |
| Slurry pit | 57 | 58 |

**Table 4.** Train and test division used in all object detectors evaluated.

| Class | Images Count | | Images % | | Instances Count | | Instances % | |
|---|---|---|---|---|---|---|---|---|
| | **Train** | **Test** | **Train** | **Test** | **Train** | **Test** | **Train** | **Test** |
| Animal | 1176 | 85 | 57.39 | 54.84 | 6542 | 193 | 66.95 | 65.20 |
| Feeder | 363 | 21 | 17.72 | 13.55 | 1730 | 39 | 17.70 | 13.18 |
| Manure pile | 100 | 6 | 4.88 | 3.87 | 669 | 16 | 6.85 | 5.41 |
| Silage ball | 346 | 35 | 16.89 | 22.58 | 765 | 40 | 7.83 | 13.51 |
| Silage S.A. | 17 | 2 | 0.83 | 1.29 | 19 | 2 | 0.19 | 0.68 |
| Slurry pit | 47 | 6 | 2.29 | 3.87 | 47 | 6 | 0.48 | 2.03 |

## 3. Results and Discussion

The objective of this work is the creation of a service to document the presence of livestock activity on a given piece of land. To achieve this, it is necessary to select the best model from the evaluated object detector algorithms, and the best deployment infrastructure.

### 3.1. Evaluation Metrics

In order to evaluate the accuracy performance of an object detector, it is necessary to define the most commonly used metrics. All of these metrics are based on:

- True Positives or TPs: number of objects detected correctly.
- False Positives or FPs: number of objects detected incorrectly.
- False Negatives or FNs: number of undetected objects.

In the object detection context, True Negatives or TNs are not taken into account because there are infinite bounding boxes that should not be detected in any image. To classify a detection as TP or FP, it is important to define what a correct detection is. If a detection does not match 100% with its respective ground truth, it does not necessarily mean that the detection is not correct. To resolve this issue, intersection over union (IOU) is used. IOU is defined as (1), and it is used to compare two regions: the bounding box produced by the detection (D) and the ground truth bounding box (G). If the IOU calculated is over a predefined threshold (the most common value in the literature is 0.5), the detection is correct (TP); otherwise, it is incorrect (FP). After computing all the TPs, FPs, and FNs, it is possible to calculate Precision and Recall. Precision, shown in (2), measures how reliable the detections are and is calculated as the percentage of correct detections (TPs) over all the detections made (TPs + FPs). Recall, shown in (3), measures the probability of ground truth objects being correctly detected and is calculated as the percentage of correct detection (TPs) over all the objects annotated in the ground truth (TPs + FNs):

$$\text{IOU} = \frac{|D \cap G|}{|D \cup G|} \tag{1}$$

$$\text{Precision} = \frac{\text{TPs}}{\text{TPs} + \text{FPs}} = \frac{\text{TPs}}{\text{Number of detections}} \tag{2}$$

$$\text{Recall} = \frac{\text{TPs}}{\text{TPs} + \text{FNs}} = \frac{\text{TPs}}{\text{Number of objects}} \tag{3}$$

A detection provided by a trained model is composed of the bounding box coordinates and a confidence score. The confidence score is used to determine how secure the model is regarding each specific detection. By using this confidence score, the detections can be sorted in descending order and the precision–recall curves for each class can be calculated. Figure 4 shows the precision–recall curve for each class obtained with YOLOv5. From these curves, the average precision, AP, can be calculated. The AP, which is represented with a number between 0 and 1, summarizes the precision–recall curve by averaging precision across different recall values. Conceptually, it represents the area under the precision–recall curve. If there is a large number of classes, it may be difficult to compare several

object detectors, as there is an AP value for each class. In this case, the mean average precision (mAP) is used. The mean average precision, which is defined in (4), is the result of averaging the AP for each class. AP and mAP has several forms of calculation [31]:

- AP(IOU = 0.5): AP with IOU threshold = 0.5 is used in PASCAL VOC [32] and measures the area under the precision–recall curve by using the all-point interpolation method.
- AP@.5 or AP@.75: are common metrics used in COCO [11] that are measured using an interpolation with 101 recall points. The difference between AP@.5 and AP@.75 is the IOU threshold used.
- AP@[.5:.05:.95]: uses the same interpolation method as AP@.5 and AP@.75, but averages the APs obtained from using ten different IOU thresholds (0.5, 0.55, ..., 0.95).
- AP Across Scales: applies AP@[.5:.05:.95] taking into consideration the size of the ground truth objects. $AP_S$ computes AP@[.5:.05:.95] with small ground truth bounding boxes ($bbox_{area} < 32^2$ pixels), $AP_S$ computes AP@[.5:.05:.95] with medium ground truth bounding boxes ($32^2$ pixels $< bbox_{area} < 96^2$ pixels) and $AP_L$ computes AP@[.5:.05:.95] with large ground truth bounding boxes ($bbox_{area} > 96^2$ pixels).

$$mAP = \frac{1}{N} \sum_{i=1}^{N} AP_i \qquad (4)$$

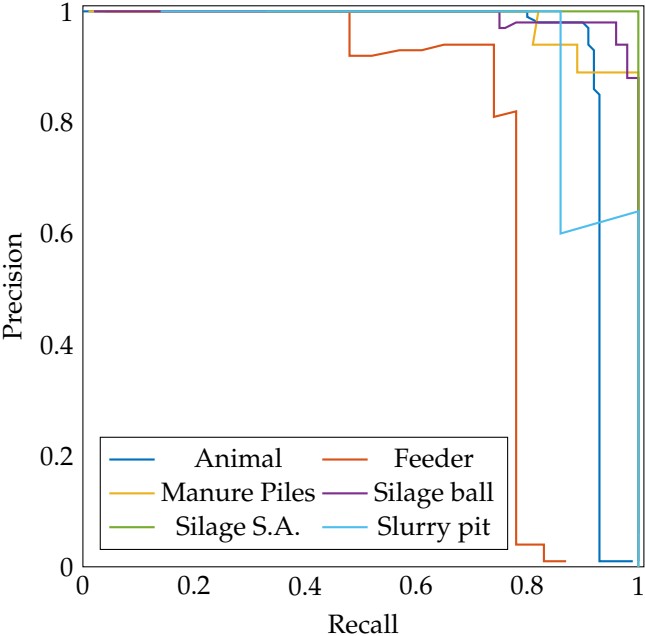

**Figure 4.** Precision–recall curves for each class using YOLOv5. The area under each curve is the AP of each class.

AP Across scales is useful when the dataset has different object sizes and aspect ratios, but as can be seen in Figure 3a,b, there is little intra-class size variability. AP@[.5:.05:.95] is a robust metric but cannot be configured to work with a fixed IOU threshold. For these reasons, in this work, AP@.5 is used to compare several object detectors. After several experiments in which hyperparameters were tuned manually, the best configurations are:

- YOLOv2:
  - Anchor boxes: five are used: ((36, 36), (152, 146), (21, 18), (68, 70), (13, 17)).
  - Backbone: Resnet50.
  - Epochs: 30.
  - Batch size: 16.
  - Learning rate: 0.0001.

- Weight decay: 0.0001.
- Solver: Adam.
- Data augmentation: image color (contrast, hue, saturation brightness), flips and scale changes.

- YOLOv4:
  - Anchor boxes: nine are used: ((13, 13), (14, 25), (22, 18), (23, 33), (40, 37), (130, 19), (61, 64), (61, 94), (123, 84)).
  - Backbone: CSPDarknet-53.
  - Epochs: 500.
  - Batch size: 32.
  - Learning rate: 0.001.
  - Weight decay: 0.0005.
  - Solver: SGD with momentum 0.937.
  - Data augmentation: image color (contrast, hue, saturation brightness), flips, scale changes and mosaic data augmentation.

- YOLOv5:
  - Anchor boxes: nine are used: ((13, 13), (14, 25), (22, 18), (23, 33), (40, 37), (130, 19), (61, 64), (61, 94), (123, 84)).
  - Backbone: CSPDarknet-53.
  - Model: S. YOLOv5 offers four models: small (S), medium (M), large (L), and extra-large (X). The larger the model, the more accurate it is, but the speed of inference is lost. In the dataset used, no improvement in accuracy is obtained by using the larger models, so small (S) is used.
  - Epochs: 500.
  - Batch size: 32.
  - Learning rate: 0.001.
  - Weight decay: 0.0005.
  - Solver: Adam.
  - Data augmentation: image color (contrast, hue, saturation brightness), flips, scale changes, and mosaic data augmentation.

- SSD:
  - Backbone: Resnet18.
  - Epochs: 40.
  - Batch size: 32.
  - Learning rate: 0.001.
  - Weight decay: 0.0005.
  - Solver: SGD with momentum 0.9.
  - Data augmentation: original SSD augmentation [8].

With these configurations, the APs per class are shown in Table 5. YOLOv5 clearly outperforms YOLOv2, SSD, and Azure Custom Vision and slightly outperforms YOLOv4; therefore, the service developed uses this object detection algorithm. It is important to highlight that the number of silage storage areas and slurry pits in the test set is too low, as can be seen in Table 4; thus, the APs obtained for these classes may not be accurate. Except for the feeder class, with YOLOv5, the AP of each class exceeds 0.90. Regarding the rest of the object detection algorithms, in the animal and manure pile class, there are no major differences between YOLOv2, SSD, and Azure Custom Vision, YOLOv2 being 2% better in the case of the animal class. With respect to the manure pile class, significant differences are obtained, Azure Custom Vision being the best performer, approaching YOLOv4 and YOLOv5 with an AP of 0.95 and followed by YOLOv2 with 0.78. SSD produces poor results for this class. The results for silage balls are similar with all detectors, although Azure Custom Vision offers the worst results with an AP of 0.79. SSD produces the best results for the silage storage area and slurry pit classes, with the exception of YOLOv4 and YOLOv5. Figure 5 shows a visual comparison among the different algorithms evaluated. Since the

objective is to find the detector that generates the best overall results, it is important to analyze the mAP. Among YOLOv2, SSD, and Azure Custom Vision, there are no significant differences, although Azure Custom Vision obtains a higher mAP than YOLOv2 and SSD. As there are not major differences between YOLOv4 and YOLOv5, the decision is given by the difference in training time, since, with YOLOv5, the training takes one hour, while, with YOLOv4, it takes three hours. In addition, the weights generated with YOLOv5 (model S) take 14 MB, while those with YOLOv4 take 421 MB. Due to this difference, which may influence the deployment, it is decided to select YOLOv5 as the object detector to carry out the livestock activity recognition service. To ensure that the partitioning of the dataset (training and testing) does not affect the results, cross testing was carried out using four subsets. The results obtained are equivalent to those shown in Table 5. Figure 6 shows some detections over the test set performed with YOLOv5.

**Table 5.** Average Precision (AP@.5) obtained for each evaluated object detection algorithm.

| Class | YOLOv2 | YOLOv4 | YOLOv5 | SSD | Azure Custom Vision |
|---|---|---|---|---|---|
| Animal | 0.84 | 0.91 | **0.92** | 0.82 | 0.82 |
| Feeder | 0.70 | 0.77 | **0.78** | 0.70 | 0.70 |
| Manure piles | 0.78 | **0.99** | 0.98 | 0.19 | 0.95 |
| Silage ball | 0.92 | **0.98** | **0.98** | 0.93 | 0.79 |
| Silage S.A. | 0.67 | **1.00** | **1.00** | 1.00 | 0.75 |
| Slurry pit | 0.71 | **1.00** | **1.00** | 0.80 | 0.79 |
| mAP | 0.770 | 0.941 | **0.943** | 0.740 | 0.800 |

The AP (and all its variants) is a good metric to compare different models, since it computes the area under the precision–recall curve regardless of the confidence obtained. However, once the model is in production, it is necessary to perform the appropriate detections using a certain confidence level. To determine the confidence threshold, it is common to use the metric known as F-Score (F1) (the harmonic mean of precision and recall) defined as (5). F1 combines precision (P) and recall (R) over a single confidence threshold. As shown in Figure 7, as confidence increases, precision increases and recall decreases. The objective is to find the confidence threshold that maximizes F1 at all classes. In this case, with a confidence of 0.727, a precision of 0.96 and a recall of 0.91 are obtained; therefore, the F1 value is 0.93. For these reasons, a confidence threshold of 0.727 is established in the tests performed. When the objective is to minimize the number of FPs, the confidence threshold should be increased. When the aim is to detect all possible objects, and the generation of PFs is not important, the confidence threshold can be lowered. It is common for an object detector to produce more than one bounding box per object; therefore, non-maximal suppression (NMS) is used to remove all redundant bounding boxes. During the NMS process, it is necessary to establish a threshold, and, in the case that the IOU among two or more predicted bounding boxes is greater than this NMS threshold, the bounding boxes with less confidence must be removed. In the dataset used, this fact is not a problem, since, as can be seen in Figure 6, object overlapping is not a problem. For this reason, the NMS threshold used is 0.5.

$$F1 = \frac{2 \times P \times R}{P + R} \tag{5}$$

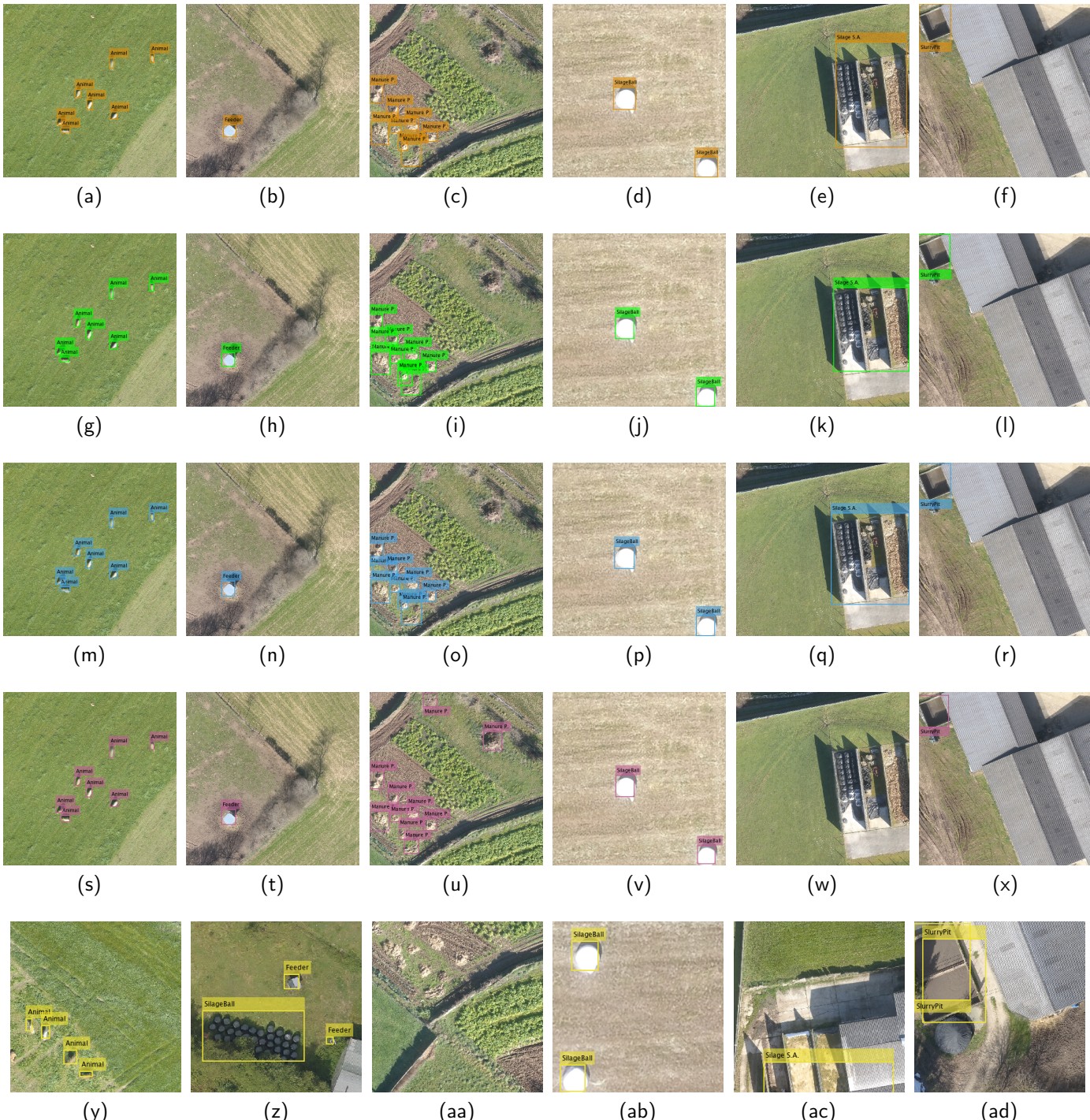

**Figure 5.** Comparison of detections performed on a subset of tests. (**a**–**f**) detections performed with YOLOv2; (**g**–**l**) detections performed with YOLOv4; (**m**–**r**) detections performed with YOLOv5; (**s**–**x**) detections performed with Azure Custom Vision; (**y**–**ad**) detections performed with SSD. The images used with SSD are not the same as those used with other detectors due to the input size required.

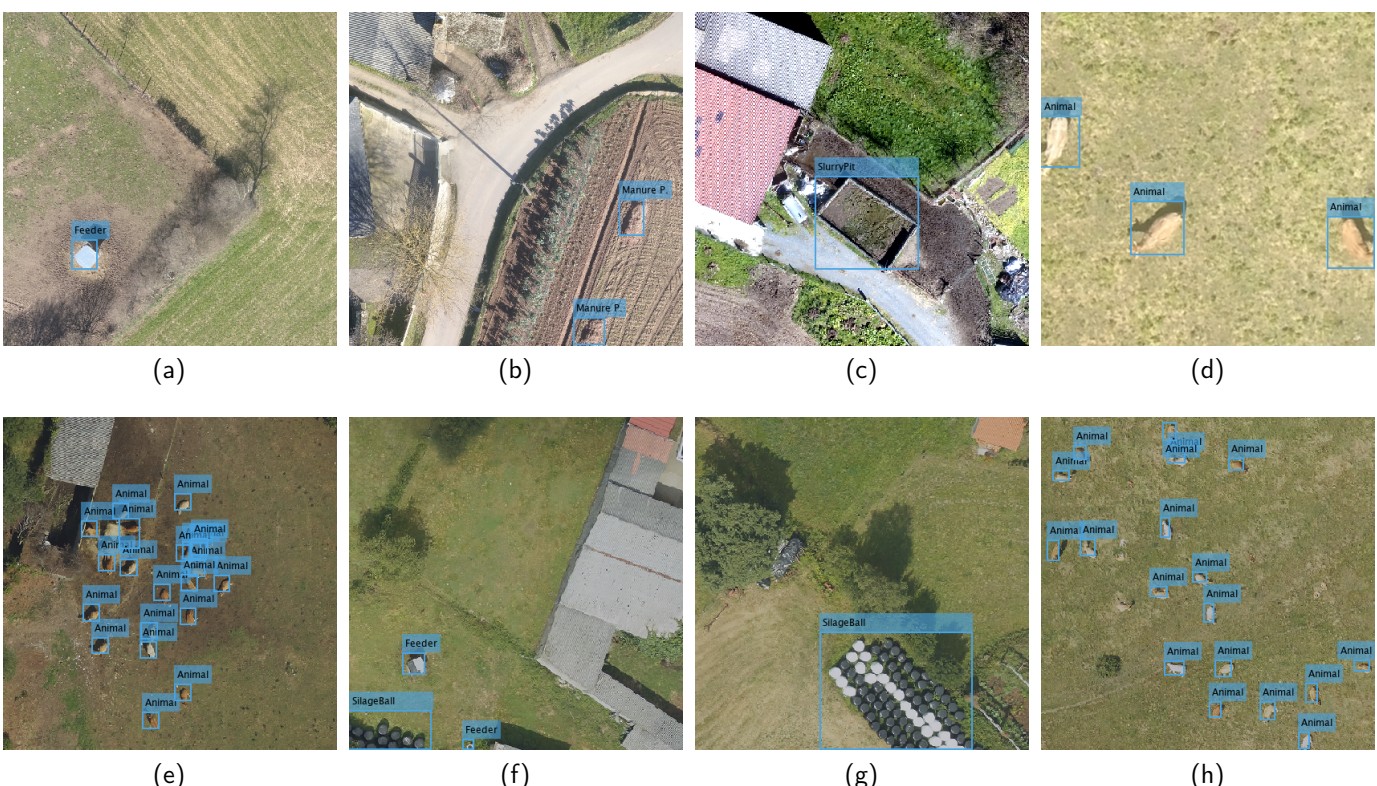

**Figure 6.** Detection results on some of the test images using YOLOv5. (**a**) one feeder; (**b**) two manure piles; (**c**) one slurry pit; (**d**) three animals; (**e**) twenty-two animals; (**f**) two feeders and one silage ball group; (**g**) one silage ball group; (**h**) eighteen animals.

### 3.2. Service Deployment

In terms of deployment infrastructure, it is necessary to analyze the system under test (SUT). The SUT is evaluated in terms of the time elapsed between the client sending a request to the server, and the client receiving the response from the server. In order to simulate the actual operation of a service, two scenarios are tested. In the first one, a batch of one hundred $416 \times 416$ images is used, and, in the second, one single $10,000 \times 10,000$ images. A local client and server are used to evaluate the SUT under same conditions as in training. This local environment is composed of two computers (client and server) with the following hardware: an Intel Core i7 9700 K CPU, 64 GB of RAM and a GeForce RTX 2080 Ti Turbo GPU. Apart from evaluating a local infrastructure, several cloud infrastructures are also evaluated for which purpose an Azure Virtual Machine is set up in the eastern US as a client and several servers in western Europe. The infrastructures evaluated in Azure as servers are classified into three types:

- IaaS: Infrastructure as a Service is a type of service where the provider rents the infrastructure and gives almost all control to the customer, making it possible to install and run a wide range of software. Normally, the cost is calculated per hour. The virtual machines evaluated are: NC6, NC12, B8MS, B12MS, D8s v3, and A8m v2.
- CaaS: Container as a Service is a form of container-based virtualization. All the set up and orchestration is given to users as a service. Azure offers GPU-based CaaS, but its support is currently limited to CUDA 9. In this case, CUDA 11 is being used, so the CaaS infrastructure has been evaluated using the full range of available CPUs (1–4) and 16 GB of RAM.
- FaaS: Function as a Service is a type of service where ephemeral containers are created. In addition, known as serverless computing, FaaS supports the development of microservices-based architectures. The cost is calculated per use.

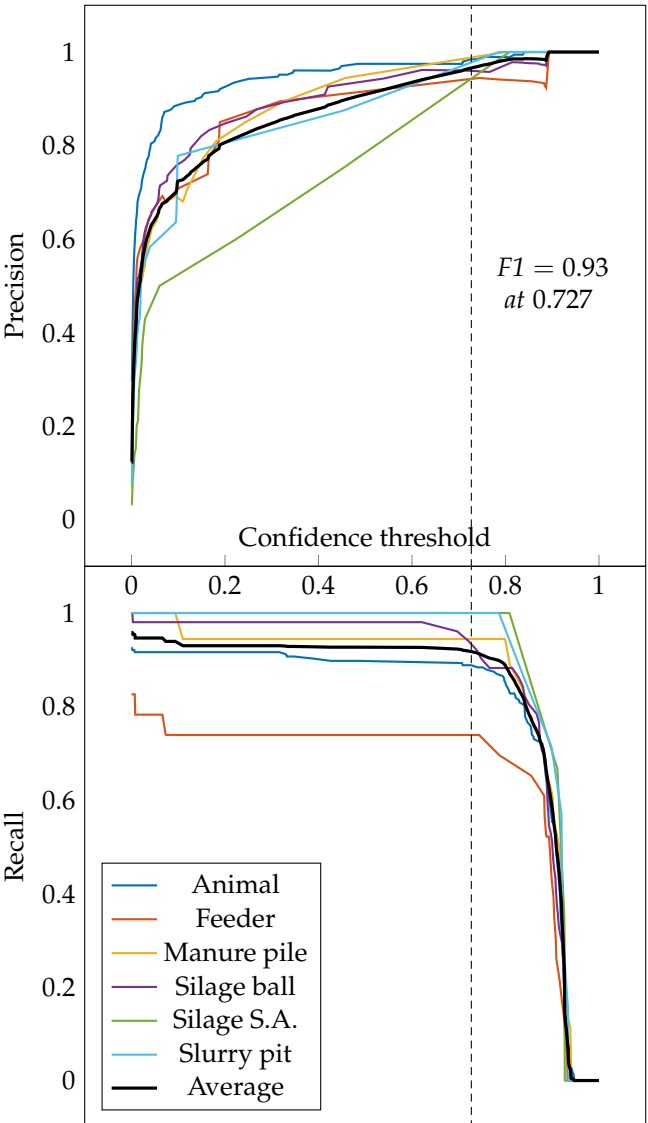

**Figure 7.** Evolution of Precision and Recall for different confidence thresholds.

Figure 8 shows the times taken in the different stages to carry out the service:

- Load time: Before starting inference, it is necessary to load the images into memory, either on CPU or GPU. This is called load time and is calculated as follows: T3 − T2 − Inference time.
- Inference time: This is the time needed to process the images in search of objects, to verify the presence of livestock activity. This time is calculated from running the detect script of YOLOv5.
- Results time: In order to confirm livestock activity, it is not enough to obtain the bounding boxes in pixels. These pixels must be translated into coordinates by creating a geojson file for each image. This translation is possible if georeferenced images are used. The time needed to do this translation is called the results time and is calculated as follows: T5 − T4.
- Input/output time: Before starting with an inference, it is necessary to save the images. Before sending the response to the client, it is necessary to zip the images with their corresponding geojsons. The sum of the two times is called input/output time and is calculated as follows: (T2 − T1) + (T6 − T5).
- Network time: The client and the server are in different locations so there is a communication time between the client and the server and vice-versa. This time

is called network time and is calculated as follows: Total time $-$ Server time $=$ $(T7 - T0) - (T6 - T1)$.

- Total time: The whole time elapsed from when the client sends a request until the server processes it and sends a response back is called the total time. It is calculated as follows: $T7 - T0$.

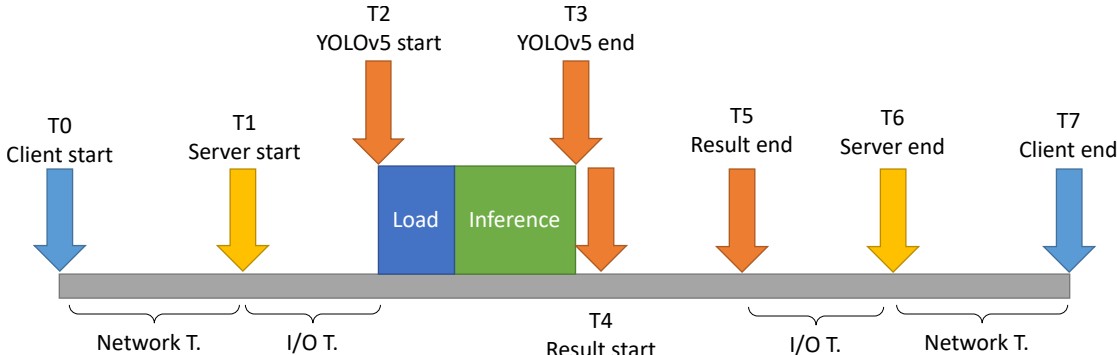

**Figure 8.** Times taken in different stages to carry out the service. Load time, inference time, results time, input/output time, and network time.

Table 6 shows the results obtained with each of the infrastructures evaluated for the first scenario (a batch of one hundred $416 \times 416$ images). Since some infrastructures do not have GPU, they were evaluated using CPU. Each infrastructure was evaluated ten times, and the different times obtained were averaged. For this reason, the standard deviation of the total time is given. For the rest of the time, the standard deviation is very small. Figure 9 shows a comparison of the different infrastructures. On top of each stacked bar is the latency ratio of each experiment calculated as (6). The latency ratio expresses the difference in time between each experiment and the fastest one.

$$\text{Latency ratio} = \frac{Experiment_i \text{ total time}}{\text{Best experiment total time}} \tag{6}$$

**Table 6.** Results of the SUT evaluation in several infrastructures using a batch of one hundred $416 \times 416$ images.

| Server | Mode | Load (s) | Inference (s) | Results (s) | I/O (s) | Network (s) | Total (s) | $\sigma_{Total}$ | Latency Ratio | Cost (€ /h) |
|---|---|---|---|---|---|---|---|---|---|---|
| Local | GPU | 2.46 | 0.64 | 9.17 | 1.45 | 0.36 | 14.08 | 0.07 | 1.0× | 0.08 |
| Local | CPU | 1.72 | 3.95 | 9.41 | 1.46 | 0.36 | 16.89 | 0.02 | 1.2× | 0.06 |
| NC6 | GPU | 4.57 | 2.45 | 12.51 | 4.11 | 2.10 | 25.75 | 0.25 | 1.8× | 0.98 |
| NC6 | CPU | 2.93 | 7.83 | 12.45 | 4.11 | 2.04 | 29.37 | 0.36 | 2.1× | 0.98 |
| NC12 | GPU | 4.88 | 2.54 | 14.48 | 4.16 | 4.13 | 30.20 | 2.65 | 2.1× | 1.97 |
| NC12 | CPU | 2.98 | 8.90 | 14.78 | 4.14 | 2.49 | 33.29 | 0.63 | 2.4× | 1.97 |
| B8MS | CPU | 4.03 | 13.41 | 15.56 | 5.14 | 11.29 | 49.44 | 4.18 | 3.5× | 0.32 |
| B12MS | CPU | 3.46 | 12.96 | 14.25 | 5.55 | 11.04 | 47.26 | 2.80 | 3.4× | 0.49 |
| D8s v3 | CPU | 3.50 | 9.02 | 14.25 | 5.31 | 11.87 | 43.95 | 2.31 | 3.1× | 0.40 |
| A8m v2 | CPU | 8.31 | 19.79 | 35.41 | 7.96 | 7.99 | 79.47 | 1.50 | 5.6× | 0.46 |
| CaaS: 1vCPU | CPU | 4.67 | 24.26 | 17.32 | 4.90 | 1.83 | 52.98 | 0.48 | 3.8× | 0.11 |
| CaaS: 2vCPU | CPU | 3.46 | 12.85 | 14.43 | 4.76 | 1.89 | 37.39 | 0.57 | 2.7× | 0.15 |
| CaaS: 3vCPU | CPU | 3.47 | 13.61 | 14.27 | 4.57 | 1.86 | 37.78 | 0.26 | 2.7× | 0.19 |
| CaaS: 4vCPU | CPU | 3.70 | 14.23 | 15.45 | 4.81 | 1.83 | 40.02 | 0.32 | 2.8× | 0.23 |
| FaaS | CPU | 2.86 | 22.02 | 7.04 | 4.25 | 12.16 | 48.33 | 1.58 | 3.4× | 0.15 |

In this first scenario, the local infrastructure has the shortest total time, see Figure 9 and Table 6. This is the expected performance as the Geforce RTX 2080 TI GPU is better than that used in Azure infrastructures (Tesla K80). It should also be borne in mind that network times have little influence on a local infrastructure. The NC6 and NC12 infrastructures have one and two GPUs, respectively. Despite having GPUs, they have also been evaluated using the CPU in order to compare them with the rest of the infrastructures. Clearly, the use of two GPUs in the case of the NC12 does not improve the inference time, and, since it costs

twice as much, it is not taken into account in the performance vs. cost comparison shown in Figure 10. B8MS, B12MS, and D8s v3 have similar performance, the only difference being that the inference time in the case of D8s v3 is lower than that obtained with B8MS and B12MS. The A8m v2 is the worst infrastructure in terms of performance, offering the worst results in load, inference, results generation, and I/O times. In the case of CaaS infrastructures, a significant improvement is obtained by moving from one CPU to two. This improvement is not obtained when replacing two CPUs with three or four because the system does not scale up. The last infrastructure evaluated is FaaS. This infrastructure obtains the best result generation time of all, but its inference time is very high. For IaaS and CaaS, the model is preloaded in the memory, so it does not affect the total service time. In the case of FaaS, it is not possible to perform this preloading because it is serverless, so this time does affect the total time. The network time is not constant, as it varies in the different infrastructures.

Performance is not the only factor to be taken into account when deploying a service: it is also important to know the cost of the infrastructure. The costs of the infrastructures used in Azure were obtained from official documentation [33–35], assuming constant requests for one hour. In the case of local infrastructure, the cost of purchasing equipment and electricity consumption are taken into account, but other costs such as maintenance, room cooling, etc. are not. The initial investment made (amortized over five years) was € 3216.66 in the case of using GPU and € 2020.66 in the case of not using it. The hourly electricity consumption is 0.074 kW with a cost of € 0.14 per kW h. This implies that the cost per hour with GPU is € 0.08 and € 0.06 without.

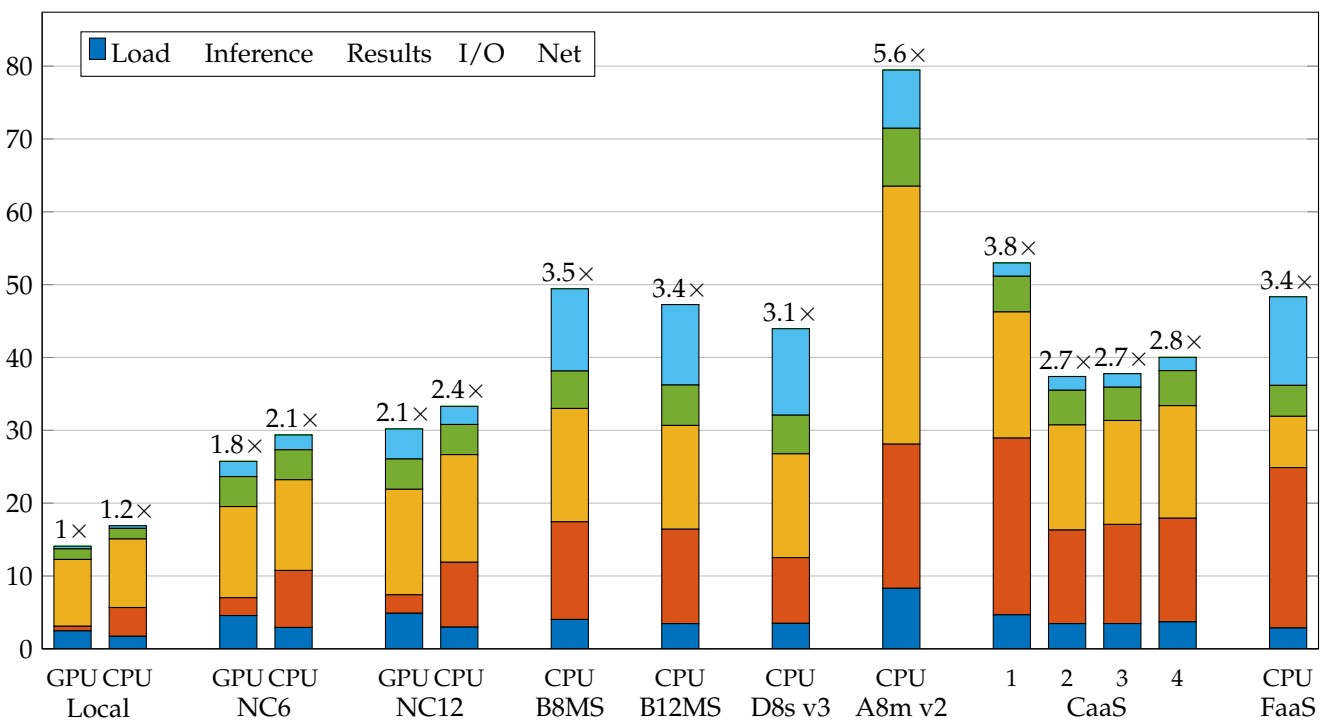

**Figure 9.** Comparison of performance offered by different infrastructures using a batch of one hundred $416 \times 416$ images. Lower values are better.

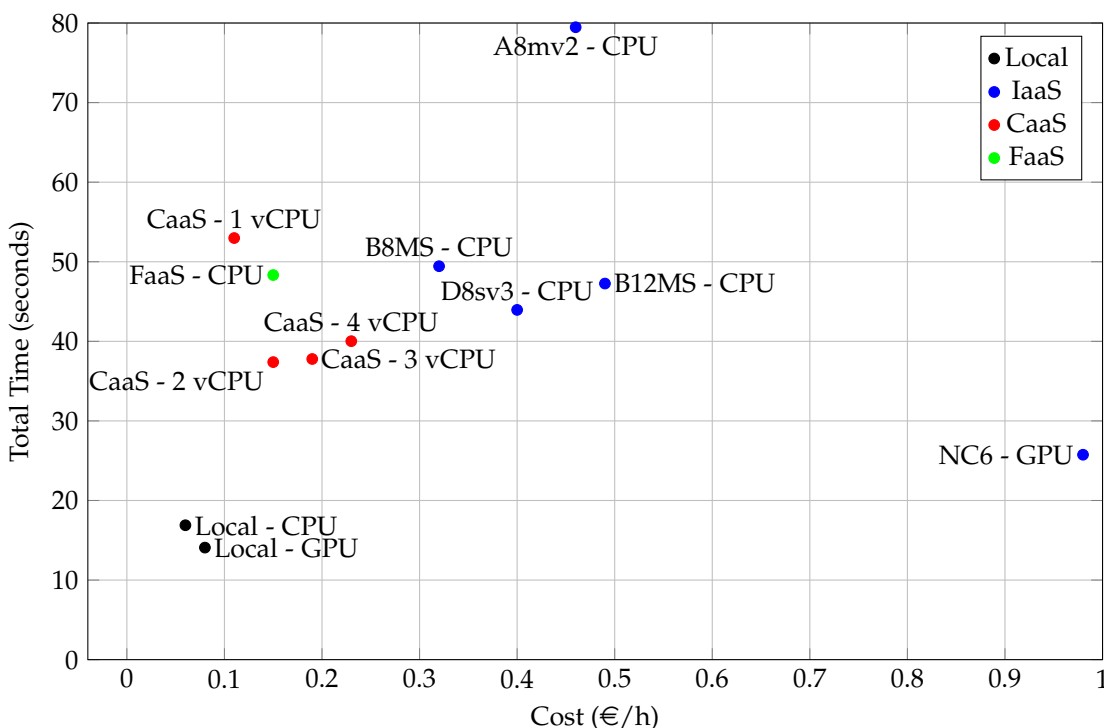

**Figure 10.** Performance vs. cost comparison using a batch of 100 images of 416 × 416 pixels. The NC12 infrastructure is not included in this chart because it performs worse than NC6 at a higher cost. NC6-CPU is not included either as in a real case it does not make sense to pay for a GPU and not use it.

Figure 10 shows a comparison of performance (total time) and cost (€/h). Clearly, local infrastructures offer the best performance/cost ratio as they are the closest to the optimal point (0, 0). The NC6 is the best performing cloud infrastructure, but, due to its high price, it does not have the best performance/cost ratio. The CaaS 2vCPU is the infrastructure that offers the best ratio due to its low cost.

In the second scenario, where a single 10,000 × 10,000 images is used for the SUT evaluation, the results are different. The local infrastructure still offers the best performance (taking into account the negligible influence of network times in local infrastructures), but, in this case, CaaS infrastructures offer lower performance than IaaS, as can be seen in Table 7 and Figure 11. The NC6 and NC12 infrastructures offer the best cloud performance, but the B8MS and B12MS also deliver good results. D8s v3 and A8m v2 infrastructures have lower performance than other IaaS. In this scenario, the FaaS infrastructure could not be evaluated, as the current maximum file size that can be received per network is 100 MB, and the image used is 400 MB. In cloud infrastructures, with the exception of the A8m v2, the load, output, I/O, and network times are similar. The most influential time is inference time, due to the size of the images to be processed. As can be seen in Table 7 and Figure 11, the result times are lower than those obtained in the first scenario (see Table 6 and Figure 9). This is due to the fact that, in the 10,000 × 10,000 images used in this scenario, 151 objects were detected (129 animals, 8 feeders, 7 silage balls and 7 slurry pits), while, in the 416 × 416 images used in the previous scenario, only 19 animals were detected. Taking into account that a batch of one hundred images (always the same image) is used, a total of 1900 animals were detected, which means that the result time is longer in the first scenario than in the second. Figure 12 shows a comparison between performance (total time) and cost (€/h) for the second scenario. Local infrastructures still offer the best performance. The B8MS and B12MS have similar results, although they are inferior in terms of performance to the NC6, and lower in cost. The A8m v2 and D8s v3 offer better

performance than the B8MS at a higher cost. Finally, CaaS infrastructures offer the lowest cost, considering their limited performance.

**Table 7.** Results of the SUT evaluation over several infrastructures using a 10,000 × 10,000 images.

| Server | Mode | Load (s) | Inference (s) | Results (s) | I/O (s) | Network (s) | Total (s) | $\sigma_{Total}$ | Latency Ratio | Cost (€/h) |
|---|---|---|---|---|---|---|---|---|---|---|
| Local | GPU | 4.56 | 0.40 | 1.08 | 2.29 | 0.30 | 8.62 | 0.13 | 1.0× | 0.08 |
| Local | CPU | 4.68 | 28.01 | 1.06 | 2.25 | 0.27 | 36.28 | 0.05 | 2.6× | 0.06 |
| NC6 | GPU | 10.15 | 4.50 | 1.81 | 16.20 | 11.22 | 43.87 | 5.84 | 3.1× | 0.98 |
| NC6 | CPU | 8.30 | 34.22 | 1.82 | 15.69 | 8.91 | 68.94 | 1.57 | 4.9× | 0.98 |
| NC12 | GPU | 10.15 | 4.48 | 1.85 | 15.86 | 15.75 | 48.10 | 7.73 | 3.4× | 1.97 |
| NC12 | CPU | 7.94 | 28.64 | 1.86 | 15.67 | 8.94 | 63.05 | 1.38 | 4.5× | 1.97 |
| B8MS | CPU | 9.59 | 36.34 | 2.25 | 16.45 | 8.15 | 72.78 | 0.80 | 5.2× | 0.32 |
| B12MS | CPU | 9.43 | 32.26 | 2.20 | 17.30 | 9.66 | 70.86 | 2.85 | 5.0× | 0.49 |
| D8s v3 | CPU | 8.66 | 52.37 | 1.99 | 16.40 | 8.06 | 87.48 | 0.74 | 6.2× | 0.40 |
| A8m v2 | CPU | 23.34 | 77.10 | 5.15 | 21.35 | 9.33 | 136.28 | 1.83 | 9.7× | 0.46 |
| CaaS: 1vCPU | CPU | 10.33 | 156.05 | 2.32 | 17.87 | 8.42 | 195.00 | 0.96 | 13.8× | 0.11 |
| CaaS: 2vCPU | CPU | 9.71 | 81.99 | 2.22 | 18.19 | 8.62 | 120.72 | 1.61 | 8.6× | 0.15 |
| CaaS: 3vCPU | CPU | 9.33 | 84.96 | 2.09 | 17.00 | 8.84 | 122.21 | 1.74 | 8.7× | 0.19 |
| CaaS: 4vCPU | CPU | 9.42 | 85.81 | 2.09 | 16.85 | 8.37 | 122.54 | 0.69 | 8.7× | 0.23 |

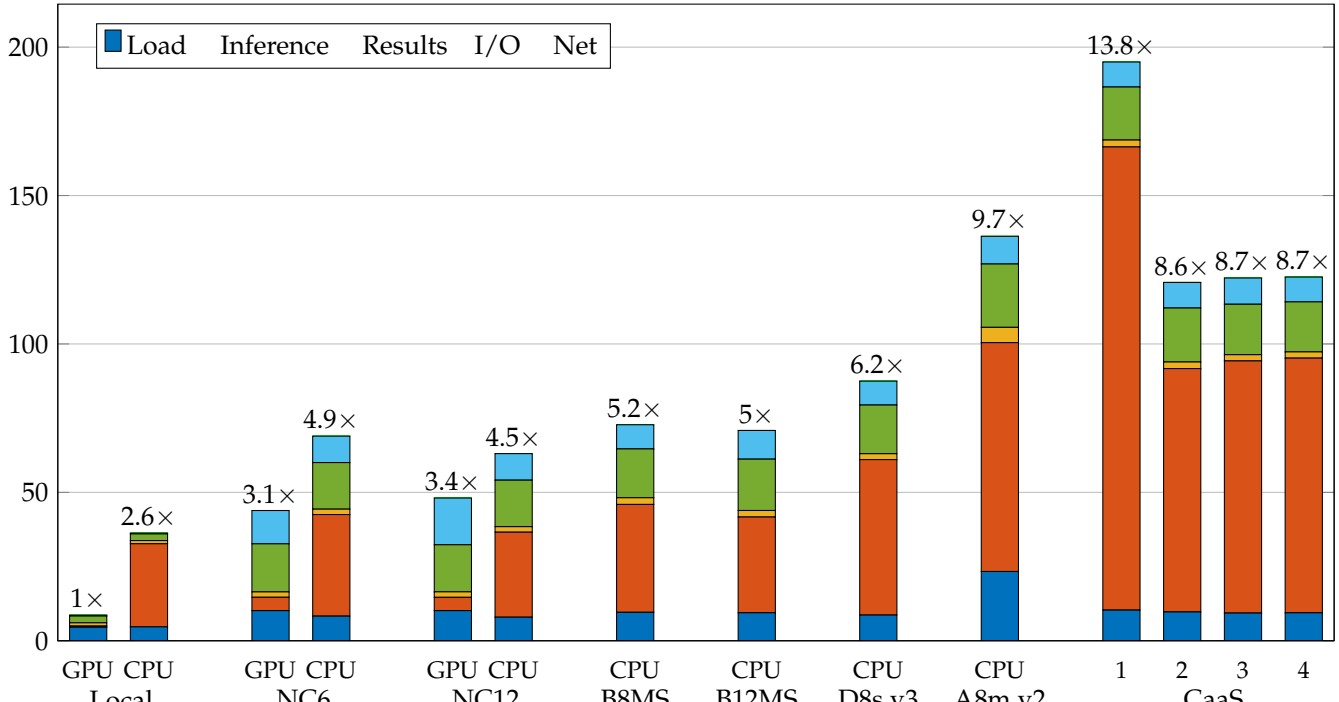

**Figure 11.** Comparison of performance offered by different infrastructures using a 10,000 × 10,000 images. Lower values are better.

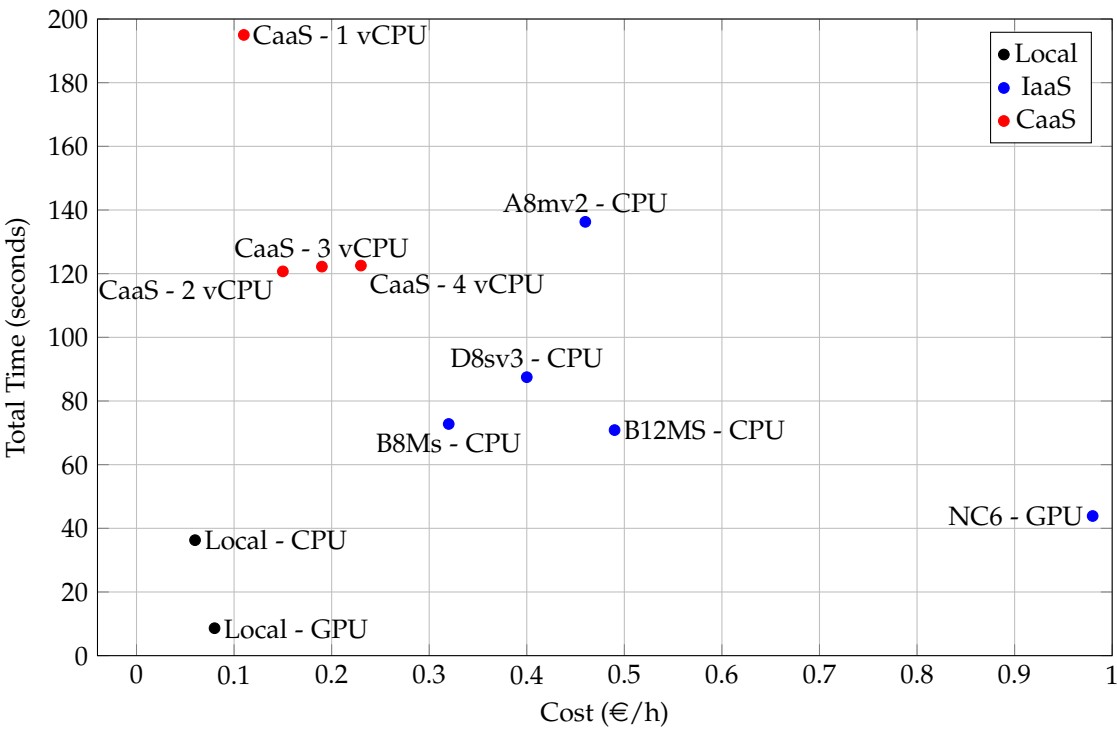

**Figure 12.** Performance vs. cost comparison using an image of $10,000 \times 10,000$ pixels. The NC12 infrastructure is not included in this chart because it performs worse than NC6 at a higher cost. NC6-CPU is not included either, as in a real case it does not make sense to pay for a GPU and not use it.

## 4. Conclusions

Recognition of livestock activity is necessary in order to apply for certain subsidies. In this paper, several object detection algorithms are analyzed to meet the recognition objectives: YOLOv2, YOLOv5, SSD, and Azure Custom Vision. The average precision metric (AP) is used to compare the detectors. Although the classes of silage storage area and slurry pit do not have a large number of annotated objects, good results are achieved with all the detectors evaluated. YOLOv5 stands out above the rest, offering an mAP of 0.94. Azure Custom Vision has an mAP of 0.80, YOLOv2 0.77, and SSD has 0.74. Due to these results, YOLOv5 is used to carry out the proposed service. When moving to a production environment, it is necessary to establish the optimal confidence threshold to perform the detections as accurately as possible. To select the most suitable threshold, the metric known as F1 was used, since it takes into account both precision and recall. As the confidence threshold 0.727 is the one that maximizes F1, it is used in all the tests performed.

The deployment of a detector to carry out this service can be accomplished by using different types of infrastructures. The local infrastructure used in this work is the one that offers the best results, but costs such as server maintenance and room cooling have not been considered. Due to the elevated additional costs that a local infrastructure can incur and the need to upgrade hardware, the current trend is to deploy services in the cloud. In this case, the infrastructure to be selected varies depending on the scenario. In the first scenario, a batch of one hundred $416 \times 416$ images is used, with the CaaS infrastructure with two vCPUs offering the best performance/cost ratio, although the NC6 infrastructure with GPU offers better performance at a higher cost. In the second scenario, where a single $10,000 \times 10,000$ pixels image is used, cloud infrastructure performance is different. The NC6 still offers the best performance, but the cost is very high. Among the other IaaS infrastructures, the B8MS has the best performance/cost ratio. The B12MS offers similar performance at a higher cost, and the D8s v3 and A8m v2 have poorer performance than

B8MS at a higher cost. The CaaS infrastructure with 2 vCPU has worse performance than the B8MS but costs less. Ultimately, the best option for implementing such a service will depend on the usage scenario and the specific requirements of the service. If a faster service is desired, the deployment can be carried out in an infrastructure with better performance, which implies a higher price. However, if the service time is less relevant, it will be more suitable to carry out the deployment in an infrastructure with worse performance but at a lower price.

Overall, this work demonstrates the feasibility of creating a service to assist in the recognition of livestock activity, an essential task to access certain subsidies. This service can be deployed in the cloud using different infrastructures depending on the requirements of the system. For future works, a dataset with more samples could be used to make the model obtained be more robust. In addition, future cloud infrastructures could be evaluated and compared with the results of this work, as well as evaluating other platforms offering cloud services.

**Author Contributions:** Conceptualization, D.G.L., O.D.P., R.U. and D.F.G.; methodology, D.G.L.; software, D.G.L.; validation, O.D.P., R.U. and D.F.G.; formal analysis, D.G.L., R.U. and D.F.G.; investigation, D.G.L. and O.D.P.; resources, R.U. and D.F.G.; data curation, D.G.L.; writing—original draft preparation, D.G.L.; writing—review and editing, D.G.L., R.U. and D.F.G.; visualization, D.G.L.; supervision, Á.A.; project administration, D.F.G.; funding acquisition, D.F.G. and Á.A. All authors have read and agreed to the published version of the manuscript.

**Funding:** This research was funded by Seresco S.A. under the contract FUO-20-018 and also by the project RTI2018-094849-B-I00 of the Spanish National Plan for Research, Development, and Innovation.

**Institutional Review Board Statement:** Not applicable.

**Informed Consent Statement:** Not applicable.

**Data Availability Statement:** Not applicable.

**Acknowledgments:** The authors would like to acknowledge Seresco S.A. for the support received, especially with the collection of the dataset, and the Spanish National Plan for Research, Development and Innovation.

**Conflicts of Interest:** The authors declare no conflict of interest.

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
