# Peer review of "Cost-Performance Evaluation of a Recognition Service of Livestock Activity Using Aerial Images"

_remotesensing, doi:10.3390/rs13122318_

Round 1
Reviewer 1 Report
Overall Decision
Minor revision
This manuscript introduces a cost-benefit evaluation of livestock activity identification service based on aerial images. Different from the existing research only on animal detection, this paper also analyzes the detection of other key elements that can be used to verify the existence of livestock activities in a given terrain. In summary, the research is interesting and provides valuable results, but the current document has several weaknesses that must be strengthened in order to obtain a documentary result that is equal to the value of the publication.
General considerations:
- At the thematic level, the proposal provides a very interesting vision, as the recognition service of livestock activity would be a very useful resource for engineers.
- The document contains a total of 33 employed references, of which 26 are publications produced in the last 5 years (79%), 6 in the last 5-10 years (18%), 1 than 10 years old (3%), implying a total percentage of 97 % recent references. In this way, the total number is sufficient, and their actuality is high.
- Technique concerns: YOLOv4 and YOLOv5 each have their own advantages. It is suggested to compare their results and choose a better algorithm.
Title, Abstract and Keywords:
- The abstract is complete and well-structured and explains the contents of the document very well. Nonetheless, the part relating to the results could provide numerical indicators obtained in the research.
Chapter 1: Introduction
- The first paragraph introducing the research topic gives a too simple view of the problems related to your topic and should be revised and completed with citations to computer vision-based authority references (Recognition and localization methods for vision-based fruit picking robots: a review).
- The novelty of the study is not apparent enough. In the introduction section, please highlight the contribution of your work by placing it in context with the work that has done previously in the same domain.
- On a general level, the study of the proposed detection techniques is reasonable, and the explanation of the objectives of the work may be valid. However, the limitations of your work are not rigorously assumed and justified.
- Vision technology applications in various engineering fields should also be introduced for a full glance at the scope of related areas. For object detection, please refer to fruit detection papers such as Three-dimensional perception of orchard banana central stock enhanced by adaptive multi-vision technology, Vision-based three-dimensional reconstruction and monitoring of large-scale steel tubular structures.
Chapter 2: The method
- Based on the complexity of the content developed in Chapter 2, you can add a flowchart to explain your case process in detail.
- In this (and later) chapters, authors are encouraged to use third and even fourth-level titles (e.g. 3.1.x or 3.1.x.x.x) for those headings that have an entity of their own. This could help to understand the structure of the manuscript's contents more easily.
Chapter 3: Experiments and results
- In this article, only the confidence threshold is determined, and the threshold of NMSalso affects the results. How do you set it?
Chapter 4: Conclusions
- It should mention the scope for further research as well as the implications/application of the study.
Reviewer 2 Report
The authors have done a brilliant job of creating a service for the recognition of livestock and farm infrastructure based on UAV images.
From a methodological point of view, given the aims of the work, there are few questions. However, there are wishes.
1. Is it possible to implement semantic segmentation rather than simple object recognition?
2. Why wasn't the UNET architecture used, which, as numerous practices have shown, also perfectly can solve the problem of automated feature detection, especially on small training sets?
3. Because of the lack of numbering in rows, it is not convenient to make remarks.
- Page 2 "Two types of datasets are used in works related to animal detection: high resolution [2,12,13] and low resolution [3,14-16]. " - resolution of what? clarify.
- page 2 "In both cases, drone (UAV)" - transcribe UAV (unmanned aerial vehicle).
-page 2 "Recognition of livestock activity. The vast majority of previous works do not address livestock activity recognition but focus only on the detection of animals, whether wild or farm animals. This article not only focuses on the presence of animals, but also evaluates other elements present on any farm: manure piles, silage balls, feeders, silage storage areas and slurry pits. These objects are proof of the presence of livestock activity." - How was the activity analyzed? Were trajectories drawn? Was the activity of each individual or the whole herd analyzed?
- Page 5 " A drone equipped with MicasenseALTUM and Alpha 7 cameras was used to collect these images. After all the images were collected, they were annotated using the following format: class_id x_center y_centerwidth height, normalizing all coordinates between 0 and 1. After this process,416×416 crops were created, obtaining the number of images and objects shown in Table 2." - Was a single orthophotoplane created or just UAV images used? If orthophotoplane, how was it obtained?
4. Why was ADAM or SGD rather than similarity metric used to reduce the loss-function in the training process? I recommend using Jacquard distance.
5. What is the spatial resolution of the original images? Page 5 - "The animal class is the largest, with 7792 instances (see Table 3), with its median for width and height being 20 and 20 pixels, respectively." 20*20 - how many meters is that?
In general, the impression of the work is positive, but it needs improvement and clarification, according to the comments.
Reviewer 3 Report
For comments, please see the attached document
